# ITERATIVE GRAPH NEURAL NETWORK ENHANCEMENT USING EXPLANATIONS

## ABSTRACT

We formulate an XAI-based model improvement approach for Graph Neural Networks (GNN) for node classification, called Explanation Enhanced Graph Learning (EEGL). The goal is to improve predictive performance using explanations. EEGL is an iterative algorithm, which starts with a learned "vanilla" GNN and repeatedly uses frequent subgraph mining to find relevant patterns in explanation subgraphs, which are then analyzed further to obtain application-dependent features corresponding to the presence of certain subgraphs in the node neighborhoods. Giving an application-dependent algorithm for such an extension of the Weisfeiler-Leman (1-WL) algorithm has been posed as an open problem. We present the results of experiments on different synthetic datasets, compare them with other feature annotations, and analyse the training dynamics.

## 1 INTRODUCTION

XAI-based model improvement is a relatively recent research direction in XAI. The general underlying observation is that explanations, besides providing information to the user about the model's decision in order to increase trustworthiness, can also be used to improve the model. Improvement can be achieved from different aspects, including predictive power, efficiency and robustness. Explanations can be used to modify the model in various ways, such as modifying the input data and its representation, the model or the loss function. A recent survey is given in (25).

In this paper we consider Graph Neural Networks (GNN) for node classification within the XAI-based model improvement paradigm. GNN form an important variant of deep neural networks for graph applications. Message-passing GNN (MPNN), the basic version of GNN, is well known to have limited representational power due to its connections to the Weisfeiler-Leman (1-WL) algorithm and the limitations of that algorithm (20; 26). For node classification problems, two nodes which are indistinguishable by 1-WL are indistinguishable by any MPNN.

Extending the power of MPNN in order to overcome these limitations is a fundamental challenge. One approach is higher-order variants of WL, which are significantly more expressive, but computationally less tractable (11). Another approach is to provide additional structural information in the form of adding *structural information as node features*. For vanilla MPNN node feature vectors are constant. The number of triangles in the 1-hop neighborhood of a node can be such an added feature. This information cannot be detected by the 1-WL algorithm. Adding node features counting the number of rooted subgraphs isomorphic to a given rooted pattern graph has been proposed in (9). They suggest domain-specific patterns, e.g., cliques for social networks and cycles for molecules. Adding new features counting the number of rooted homomorphisms is proposed in (8). Building on these proposals, a general question, formulated in (8), is how to select useful patterns? We refer to this problem as the *GNN pattern selection problem*. It is noted in (8) that "one important remark is that selecting the best set of features is still a challenging endeavor". It is also pointed out that the *application-dependence* of the best set of features adds to the difficulty of the problem.

XAI-based model improvement suggests to use explanations for solving the GNN pattern selection problem. Explanations are often of the form of a *rooted subgraph*.

We propose the EXPLANATION ENHANCED GRAPH LEARNING (EEGL) approach for iterative enhancement of the predictive performance of GNN for node classification. EEGL uses *frequent connected subgraph mining* to identify patterns which occur in many explanation subgraphs for

nodes of a given class. These patterns are then analyzed further to select the best patterns. Features to be introduced represent the presence or absence of these patterns in the neighborhood. Starting with vanilla MPNN, the process can be iterated. We use *synthetic* data with *ground truth explanations*. This choice is justified in detail in Section 4.1.

A few remarks are in order about assumptions underlying the approach and its intended scope of applications [1]. It is assumed that the learning problem is such that 1. *there exist explanation subgraphs of reasonable quality* and 2. *the GNN learning and explanation algorithm to be improved is of reasonable quality*. The *hypothesis*, then, is that if assumptions 1. and 2. hold then explanations can be used to construct useful features, and those can be used to improve performance. Assumptions 1. and 2. are *not* expected to hold in general. On the other hand, in view of assumption 2., the goal is *to improve a weak learner using weak explanations*

Applications where the assumptions *may* hold could be in *scientific discovery*. In this context, EEGL may be thought of as describing the following scenario (related to previous work on XAI and the philosophy of science, such as (21)) . A scientist has an idea about some kind of explanation of a property. This can be communicated to the data scientist, and running EEGL may be helpful in exploring, and perhaps correcting, the initial idea. If EEGL fails, then this *may* be an indication that the original idea was not correct.

For the experimental evaluation of the EEGL approach, one can consider the following questions:

Q1: Can the performance of GNNs be improved by using frequent subgraph mining of explanations? Do iterations help?

Q2: How does the performance of EEGL-trained GNN compare to other feature initializations?

Q3: What is the effect of the relationship between motif symmetries and label partitions?

The objective of the paper is to provide first steps towards answering these questions. *Regarding Q1 and Q2 the overall results are positive.* For Q2 we considered three other feature assignments: 1. one-hot encoding of true class labels, 2. randomly assigned numerical features, 3."maliciously" selected subgraph features. Here 2. is related to results of (1; 10) on the power of random initialization and 3. to the comment of (8) that performance "almost always benefits from any set of additional features". Q3 is one of the main research questions concerning overcoming the weaknesses of MPNN. We started with benign cases towards understanding the viability of the approach.

**Contributions** In summary, the contributions of the paper are the following: it (i) applies an XAI-based iterative model improvement method to GNN, (ii) provides an automated, application-dependent solution to the GNN pattern selection problem, and (iii) applies frequent connected subgraph mining to XAI.

**Outline** The rest of the paper is organized as follows. In Sections 2 we overview related work. The EEGL system is presented in Section 3. We report and discuss the experimental results obtained in Section 4. Finally, in Section 5 we conclude and formulate some problems for further research.

## 2 RELATED WORK

Besides the most closely related works, already cited in Section 1, we mention briefly some further references, using survey papers covering related work when possible. A recent general survey of XAI, including some discussion of the role of XAI-based model improvement, is (6). In the survey of (19) of the different variations of the Weisfeiler-Leman algorithm, Section 5.2 is about neural architectures extending 1-WL, including (8; 9). A subsection discusses other subgraph-enhanced approaches, including (29) on node classification, which implement some form of symmetry-breaking to address the limitations of 1-WL. Thus these approaches exploit properties of the network, but do *not* take the learning problem into consideration. A detailed computational study is given in (11), including results on $k$-WL. The need for synthetic examples is discussed in (18).

Explanatory interactive machine learning (13; 14) considers explanations returned by the learner and corrected by the user as part of the human-in-the-loop learning process. Faithfulness, a basic metric

---

[1]See also (25) for related remarks on XAI-based model improvement in general.

---

**Algorithm 1** EXPLANATION ENHANCED GRAPH LEARNING (EEGL)

---

**Input:** graph $G$, set $T \subseteq V(G) \times C$ of training examples for some finite set $C$ of class labels, node feature vector dimension $d$, relative frequency threshold $\tau \in (0, 1]$, and iteration number $K$
**Output:** GNN model $\Phi : V(G) \to C$

1: $X \leftarrow$ INIT_FEATURE_MATRIX$(G, d)$
2: **for** $k = 1, \dots, K$ **do**
3: $\quad \Phi \leftarrow$ GNN_LEARNING$(G, X, T)$ $\qquad\qquad\qquad$ ▷ GNN Learning Module
4: $\quad$ **for all** $c \in C$ **do** $\mathcal{E}_c \leftarrow \emptyset$ $\qquad\qquad\qquad$ ▷ Node Explainer Module
5: $\quad$ **for all** $v \in V(G)$
6: $\qquad E_v \leftarrow$ GNN_NODE_EXPLAINER$(G, X, \Phi, v)$
7: $\qquad$ add $E_v$ to $\mathcal{E}_c$, where $c = \Phi(v)$
8: $\quad$ **for all** $c \in C$ **do** $\qquad\qquad\qquad\qquad\qquad$ ▷ Pattern Extraction Module
9: $\qquad \mathcal{P}_c \leftarrow$ MAXIMAL_FREQUENT_PATTERN_MINING$(\mathcal{E}_c, \tau)$
10: $\quad \mathcal{P}^\top \leftarrow$ TOP_ROOTED_PATTERNS$(\mathcal{P}, \min\{d, |\mathcal{P}|\}, T)$, where $\mathcal{P} = \bigcup_{c \in C} \mathcal{P}_c$
11: $\quad X \leftarrow$ UPDATE_FEATURE_MATRIX$(G, \Phi, \mathcal{P}^\top, d)$ $\qquad$ ▷ Feature Annotation Module
12: **return** $\Phi$

---

for evaluating explanations, measures the predictive power of explanation subgraphs (4) [2]. Other works, closer to the present paper, use explanations in an automated manner as a tool for improving prediction. In (5) the effect of incorporating linear approximations into the learning process is considered. Similarly, (3) uses explainability information to guide message passing in GNN. The SUGAR system (24) uses a reinforcement pooling mechanism to incorporate significant subgraphs into graph classification.

## 3 THE EEGL SYSTEM

In this section we present the main components of the EXPLANATION ENHANCED GRAPH LEARN-ING (EEGL) system. Background on GNN, GNNEXPLAINER, the Weisfeiler-Leman algorithm, and frequent subgraph mining is given in (15; 23; 27) and Appendix A. For a graph $G$, $V(G)$ and $E(G)$ denote the sets of nodes and edges. Graphs are always undirected. A *rooted graph* is a pair $(G, v)$, where $v \in V(G)$. Graph $G_1$ is *isomorphic* to graph $G_2$ if there is a *bijection* $\psi : V(G_1) \to V(G_2)$ such that $\{u, v\} \in E(G_1)$ iff $\{\psi(u), \psi(v)\} \in E(G_2)$ for all $u, v \in V(G_1)$. Furthermore, $G_1$ is *sub-graph* (resp. *induced subgraph*) *isomorphic* to $G_2$ if $G_2$ has a subgraph (resp. induced subgraph) that is isomorphic to $G_1$, or equivalently, if there exists an *injective* function $\varphi : V(G_1) \to V(G_2)$ such that $\{\varphi(u), \varphi(v)\} \in E(G_2)$ if $\{u, v\} \in E(G_1)$ (resp. $\{\varphi(u), \varphi(v)\} \in E(G_2)$ iff $\{u, v\} \in E(G_1)$), for all $u, v \in V(G_1)$). A rooted graph $(G_1, r)$ has a *rooted subgraph isomorphism* (resp. *rooted induced subgraph isomorphism*) into a rooted graph $(G_2, v)$ if there is a subgraph isomorphism (resp. induced subgraph isomorphism) $\varphi$ from $G_1$ into $G_2$ such that $\varphi(r) = v$. A *rooted pattern* is a pair $(P, v)$ such that $P$ is a connected graph and $v \in V(P)$.

The pseudocode of EEGL is given in Alg. 1 (see, also, Fig. 4 in Appendix B for a high-level depiction of the EEGL process). It consists of the (i) GNN learning, (ii) node explainer, (iii) pattern extraction, and (iv) feature annotation modules. The four modules are used to learn an unknown target function $f : V(G) \to C$, where $G$ is the input graph and $C$ is a finite set of class labels. In addition to $G$, the input to EEGL contains also a set $T = \{(v, f(v)) : v \in V'\}$ of training examples for some $V' \subseteq V(G)$, the dimension $d$ of the node feature vectors, a relative frequency threshold $\tau \in (0, 1]$, and a positive integer $K$ specifying the number of iterations (cf. Alg. 1). In each iteration of the outer loop in Alg. 1 (lines 2–11), all nodes of $G$ are associated with a $d$-dimensional *feature vector*. The feature vectors of the nodes are represented by an $n \times d$ *feature matrix* $X$, where $n = |V(G)|$. We now describe the above four modules.

**GNN Learning Module (line 3 of Alg. 1)** In each iteration, EEGL first learns a new GNN $\Phi$ for $G$ using $G$, $X$, and $T$ as input. While $G$ and $T$ are never changed, $X$ is recalculated in each iteration.

---

[2] A comment on terminology. In this paper we do not discuss the evaluation of explanations. Thus we mean accuracy in the standard ML, and not in the XAI sense, as used in (4) and somewhat differently in (27).

It is initialized in line 1 for the first iteration and updated in line 11 for the further iterations. We use the "vanilla" initialization, i.e., $X$ is initialized with the $n \times d$ matrix of ones. The function GNN_LEARNING (line 3) is realized with graph convolutional networks (GCN) (17).

**Node Explainer Module (lines 4–7 of Alg. 1)** The GNN model $\Phi$ is used as input to the *node explainer* module, together with $G$ and $X$. This is called for each $v \in V(G)$ separately (lines 5–6). It returns an *individual* subgraph $P_v$ of $G$ as the explanation for the model's prediction of the class of $v$ by $\Phi(v)$. In our experiments, the node explainer function GNN_NODE_EXPLAINER (line 6) is realized with the GNNEXPLAINER system (27). Most explanation graphs returned by GNNEXPLAINER contained the nodes themselves. This property is crucial for EEGL.[3] If $v \in P_v$, it is marked as the *root* of $P_v$. We note that GNNEXPLAINER calculates also a feature mask for each explanation pattern. These are disregarded in EEGL, but could play a role in future work. The explanation graphs are partitioned according to the corresponding *predicted* class labels (lines 4 and 7); the block containing the explanation graphs for a class $c \in C$ is denoted by $\mathcal{E}_c$ (cf. line 7).

**Pattern Extraction Module (lines 8–10)** This module generates a set of *maximal frequent rooted* patterns from the explanation graphs computed by the previous module. This will then be used by the next module. The underlying assumption behind the synthetic examples is that there is a set of *class patterns* for each class label. More precisely, for input graph $G$, target function $f$, and class label $c \in C$ there is a set $S_c$ of (almost) *contrastive* rooted patterns such that (i) for most [4] $v \in V(G)$ with $f(v) = c$, there is a rooted pattern $(P_c, r_c) \in S_c$ and a rooted subgraph isomorphism from $(P_c, r_c)$ to $(G, v)$ and (ii) there is *no* such rooted pattern in $S_c$ and rooted subgraph isomorphism for *most* $v' \in V(G)$ with $f(v') \neq c$. The set $S_c$ is the underlying *ground truth*.

Let $v \in V(G)$ be a node selected uniformly at random such that $\Phi(v) = c$ for some $c \in C$ and let $(P_c, r_c)$ be a rooted pattern from $S_c$. It follows from the assumptions that with a certain probability, a rooted explanation graph $(P_v, v) \in \mathcal{E}_c$ computed for $v$ contains a subgraph $P'_v$ such that $v \in V(P'_v)$ and $(P'_v, v)$ is a rooted subgraph of $(P_c, r_c)$ (i.e., there is a rooted subgraph isomorphism from $(P'_v, v)$ to $(P_c, r_c)$). Since $P'_v$ is a subgraph of $P_v$ and rooted subgraph isomorphism is used for pattern matching, $(P'_v, v)$ contains *less* structural constraints. Hence, it can be regarded as a *generalization* of $(P_v, v)$.[5] Thus we need to calculate a set of rooted patterns that generalize a large fraction of the rooted explanation graphs in $\mathcal{E}_c$ and, in order to avoid redundancy, are *most specific* at the same time with respect to this property. As mentioned above, there are two sources of errors for the explanation patterns computed by the previous module. First, with a certain probability, the true class label $f(v)$ is predicted *incorrectly* by $\Phi$ (i.e., $f(v) \neq c = \Phi(v)$). Second, there is *no* guarantee that the explanation graph $(P_v, v)$ provides a *genuine* explanation for predicting the class label of $v$ by $\Phi(v) = c$, independently whether or not $\Phi(v) = f(v)$. Still, it is reasonable to assume that many of the *individual* explanation patterns computed for $c$ have a relatively large overlap with some of the unknown class patterns in $S_c$. Accordingly, we expect that *most frequent* rooted subgraphs of $\mathcal{E}_c$ computed in the node explanation module are actually rooted *subgraphs* of some patterns in $S_c$.

The above arguments motivate considering the *maximal frequent* rooted subgraphs of $\mathcal{E}_c$ (i.e., which are frequent and all their proper connected supergraphs are infrequent in $\mathcal{E}_c$ w.r.t. $\tau$) (line 9). The assumptions imply that they contain at least a part of the structural information assigning a node to class $c$. These patterns can be regarded as the *most specific generalizations* of most of the individual explanations in $\mathcal{E}_c$. ( (For more details please see Appendix C.)

After the computation of the maximal frequent rooted subgraphs for all $c \in C$ (lines 8–9), a small subset $\mathcal{P}^\top$ of *top* rooted patterns is selected from $\mathcal{P} = \bigcup_{c \in C} \mathcal{P}_c$ (line 10 ). More precisely, for each $(P, r) \in \mathcal{P}_c$, EEGL evaluates how well $(P, r)$ performs as a classifier for class $c$ by computing its F1-score on the training set $T$. The module then returns the top $d'$ patterns with the highest F1-scores in a round-robin fashion from $\mathcal{P}$, where $d' = \min\{d, |\mathcal{P}|\}$.

**Feature Annotation Module (line 11)** Using $\mathcal{P}^\top = \{(P_1, r_1), \ldots, (P_{d'}, r_{d'})\}$, in the feature annotation module we update $X$ for the next iteration of the main loop by setting $\vec{x}_v[j]$ to 1 if there is a rooted subgraph isomorphism from $(P_j, r_j)$ to $(G, v)$; otherwise to 0, for all $v \in V(G)$ and $j \in [d']$. If $d' < d$, the last $d - d'$ entries in $\vec{x}_i$ are set to some value.

---

[3]Parameters can be chosen so that GNNEXPLAINER (27) is forced to include $v$ in $P_v$ for *all* $v \in V(G)$.

[4]The qualifications *almost, most* refer to both noise in the data and errors in prediction and explanation.

[5]By generalization we mean the relationship between two rooted patterns in the poset of all rooted patterns defined by rooted subgraph isomorphism, and *not* "generalization" from data as used in machine learning.

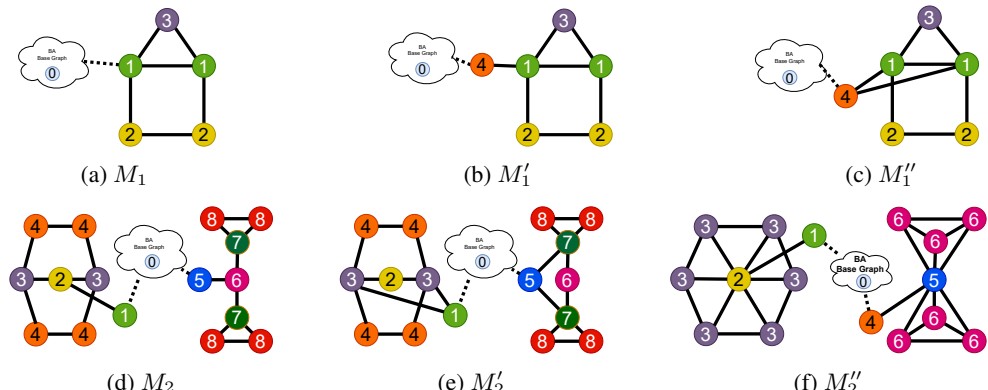

(a) $M_1$        (b) $M_1'$        (c) $M_1''$

(d) $M_2$        (e) $M_2'$        (f) $M_2''$

Figure 1: The motifs used in the dataset generation: The "house" motif (a) with asymmetric (b) and symmetric extensions (c), motif pairs with 1-WL indistinguishable nodes (d,e) and a supergraph of (d) keeping the nodes 1-WL indistinguishable (f).

Experiments with EEGL as described above showed that its runtime is infeasible on larger graphs, the culprit being the annotation module. To speed up EEGL, we therefore apply two heuristics in the feature annotation module. First, we use rooted *induced* subgraph isomorphism, instead of rooted subgraph isomorphism. Second, when deciding if there exists a rooted induced subgraph isomorphism from a rooted pattern $(P, r)$ into $(G, v)$ for some $v \in V(G)$, it suffices to consider the $\epsilon$-hop neighborhood of $v$, where $\epsilon$ is the *eccentricity* of $r$ in $P$ (i.e., the maximum distance between $r$ and any other node $u \in V(P)$). Instead of $\epsilon$, we use the *radius* $\rho$ of $P$ (i.e., the *minimum* of the node eccentricities over all nodes in $P$). Feature vector annotation using these heuristics is sound, but incomplete (i.e., it cannot return false positives). Still, our experimental results reported in the next section clearly demonstrate that even with these heuristics, EEGL is able to achieve significant improvements in predictive performance, in practically feasible time.

## 4 EXPERIMENTAL EVALUATION

In this section we experimentally evaluate the EEGL system on synthetic data and address the questions formulated in the introduction. In Section 4.1 we describe the datasets used and the experimental setup. The description of the datasets is accompanied by discussions of the combinatorial considerations behind the choice of the motifs. In Section 4.2 these considerations are matched by discussions on how they explain different observations on the performance of the extended 1-WL algorithm (both successes and failures).

### 4.1 DATASETS AND EXPERIMENTAL SETUP

**Synthetic Data** Exploring the possibilities and limitations of EEGL is a long-term project. The first "sanity check" is a pilot project in the most favorable circumstances. That is provided by the kind of *synthetic* examples we consider, with ground truth available. A detailed understanding of this case is a challenging task in itself, and that is the topic of this first paper on the approach. The use of synthetic data is also justified by (18), discussing the SOTA on node classification benchmarks. According to (18), current real-world datasets are *unsuitable* for a deeper understanding as they do not allow for a systematic study of the effect of various graph parameters on performance. The focus of (18) is on parameters such as class size distributions, and edge densities between classes (corresponding to heterophily versus homophily). We are interested in the effect of the *structural complexity* of the node classification task, which can be relevant, for example, for scientific applications. We focus on the approach of starting with the synthetic model of BA-graphs with motifs attached. This model has been introduced in (27) and it has become a standard benchmark for GNN. We gradually increase structural complexity by changing the complexity of the motifs attached.

**Datasets** Synthetic graphs used in the experiments are constructed as follows. We first generate a Barabási-Albert (BA) random base graph (7), and then attach copies of one or two small graph motifs to $m$ nodes selected uniformly at random from the base graph. The motifs used in our experiments

are shown in Fig. 1, together with specifying how to attach them to the base graph. As an example, the "house" motif (Fig. 1a) used in GNNEXPLAINER (27) is attached to the base graph via one of its nodes with label 1. For all motifs we use the same node and motif numbers as in (27), i.e., $n = 300$ and $m = 80$. For $M_2, M_2', M_2''$, we select one of the two motifs uniformly at random and attach it to the base graph. Thus, the size of the graphs is between 700 and 940. In the next step we assign a class label to each node of the graph obtained. In particular, the class labels of the motifs' nodes are indicated in Fig. 1, while the nodes of the base graph are all labeled by 0. Finally, we add a small amount of *structural noise* to the motifs by selecting a small set of random edges.

Recall that our underlying assumption behind the data is that each node class can be characterized with a set of (unknown) contrastive rooted graph patterns. To see that the graphs in the experiments fulfill this assumption, note that for each motif in Fig. 1, all nodes having the same label can be distinguished from the other nodes by some characteristic rooted pattern(s). For example, the node with label 3 in Fig. 1a can be distinguished from the nodes with label 1 or 2 by the "house" graph with the top node as the root. Regarding the nodes of the base graph, our experimental results suggest that most of them can be represented by a set of rooted patterns. We cannot expect a clear contrastive property for *all* classes because of the structural noise added in the last step to the graph. We note that the class of the base graph nodes is handled in the same way as all other node classes (i.e., EEGL has no prior knowledge of the graph generation process).

The following considerations motivated our choice of the motifs in Fig. 1. Regarding the "house" graph motif $M_1$ (Fig. 1a), its nodes belong to three different classes. Note that without the dashed edge attaching $M_1$ to the base graph, the three labels indicate the *node orbits*[6] of this motif, i.e., the equivalence classes of the nodes under automorphisms. Note also that by attaching $M_1$ to the base graph via *one* of its nodes labeled by 1, we implicitly break its inherent symmetry, obtaining five singleton node orbits. This motif is considered among others for a "historical" reason; it was used in the experimental evaluation of GNNEXPLAINER (27) and also by the recent graph data generator SHAPEGGEN (3) [7]. The second motif $M_1'$ (Fig. 1b) is obtained from $M_1$ by adding a new edge to it. Its endpoint labeled by 4 has a strong impact on the node orbits: It breaks the symmetry of the "house" graph explicitly, and *not* via the attachment to the base graph. In particular, the six nodes of $M_1'$ belong to six different (singleton) node orbits. One can easily check that they are pairwise distinguishable by 1-WL. Thus, the class labels in case of this motif do not reflect the motif's node orbits and 1-WL labels. The reason of considering this motif is to investigate whether the GNN trained via EEGL is still able to handle this kind of "merged" node classes. In $M_1''$ (Fig. 1c), the third version of the "house" motif, we attach the motif to the base graph via a node connected to *both* nodes of label 1. The symmetry of the "house" graph is *not* destroyed in this way. In particular, the class labels represent the node orbits. Note, however, that two nodes of the same label that belong to *different* occurrences of the motif in the graph can have different 1-WL labels because of asymmetries in the whole graph. Our purpose with this motif is to study whether the iteratively upgraded GNN is able to ignore structural information making locally (i.e., w.r.t. the motif only) indistinguishable nodes globally distinguishable.

The other three motifs in Fig. 1 are motif pairs, with class labels indicating their node orbits. A reason for considering them is that certain nodes in the motif pairs are indistinguishable by 1-WL. For example, in case of $M_2$ (Fig. 1d), nodes with label 4 have the same 1-WL label as those with label 8. We have a similar situation for labels 1 and 5, 2 and 6, and 3 and 7. Since the corresponding nodes are indistinguishable by 1-WL, they are indistinguishable by GNN. In case of $M_2$ we have, however a situation similar to $M_1''$ (Fig. 1c). Being attached to different base graph nodes, locally indistinguishable nodes can become distinguishable in the global graph. Thus, one of our goals with $M_2$ and its variants $M_2'$ (Fig. 1e) and $M_2''$ (Fig. 1f) is to examine the influence of the asymmetry in the base graph on the predictive performance of EEGL. There are only small structural differences between $M_2$ and $M_2'$ and between $M_2$ and $M_2''$. While $M_2'$ is obtained from $M_2$ by edge deletion and insertion, $M_2''$ is a supergraph of $M_2$. Our goal with $M_2'$ and $M_2''$ is to study not only the issue discussed for $M_2$, but also the sensitivity of EEGL to structural changes.

---

[6]Two nodes $u, v$ belong to the same node orbit iff there is an automorphism (i.e., an isomorphism from the motif to itself) mapping $u$ to $v$. If two nodes are in the same node *orbit* then their 1-WL labels are identical. The converse of this claim is, however, not true.

[7]This paper appeared just before the submission of our paper. In future work we are going to evaluate EEGL on other graphs generated by SHAPEGGEN as well.

| | $M_1$ | $M_1'$ | $M_1''$ | $M_2$ | $M_2'$ | $M_2''$ |
|---|---|---|---|---|---|---|
| LE | $100.0 \pm 0.00$ | $99.86 \pm 0.46$ | $100.0 \pm 0.00$ | $99.78 \pm 0.46$ | $99.15 \pm 0.83$ | $99.58 \pm 0.54$ |
| A | $98.42 \pm 1.60$ | $97.16 \pm 2.11$ | $98.71 \pm 1.58$ | $51.01 \pm 7.77$ | $52.48 \pm 7.06$ | $65.13 \pm 7.57$ |
| R | $97.57 \pm 1.37$ | $92.76 \pm 4.15$ | $98.98 \pm 0.99$ | $80.08 \pm 4.07$ | $79.44 \pm 5.41$ | $\mathbf{93.45 \pm 4.89}$ |
| R0 | $98.28 \pm 1.64$ | $90.19 \pm 5.80$ | $\mathbf{99.57 \pm 0.69}$ | $51.38 \pm 7.06$ | $52.24 \pm 6.36$ | $55.14 \pm 4.60$ |
| R1 | $\mathbf{99.71 \pm 0.61}$ | $99.44 \pm 0.96$ | $99.43 \pm 1.00$ | $89.36 \pm 18.19$ | $87.91 \pm 14.59$ | $78.02 \pm 16.70$ |
| R2 | $\mathbf{99.71 \pm 0.61}$ | $\mathbf{99.86 \pm 0.44}$ | $99.14 \pm 1.38$ | $\mathbf{91.83 \pm 14.04}$ | $\mathbf{98.11 \pm 1.78}$ | $84.02 \pm 14.04$ |

Table 1: Average weighted F1-score results in percentage (mean $\pm$ standard deviation) obtained with 10-fold cross-validation for the motifs in Fig. 1 with the label encoding (LE), adversarial (A), random (R) settings, and for the three iterations of EEGL (R0, R1, R2).

**Experimental Setup** We now describe the method for evaluating the performance of EEGL. Recall from Section 3 that the input graph $G$ is associated with a feature matrix $X$ of size $n \times d$, representing the feature vectors of the $n$ nodes of $G$. For each graph $G$ used in the experimental evaluation, we have carried out four experiments, three with GCN (17) (used also in EEGL) and one with our EEGL system. The GCN experiments use precomputed feature vectors. Motivation for choosing these settings and discussion of the experiments are given in the next section.

In the *label encoding* setting, feature vectors are essentially one-hot encodings of the node labels. In the *adversarial* setting, feature annotation is done with a precomputed set of rooted patterns. These do not have any rooted subgraph isomorphism into the motif nodes, but do have rooted subgraph isomorphisms to some nodes in the base graph. In the *randomized* setting, feature annotation is done using precomputed random numbers from $[0, 1]$.

Regarding the parameters, in case of $d$ we follow (27) and set it to 10 in all experiments. EEGL calls GASTON (22) to generate the candidate frequent patterns. This subroutine has two parameters, a frequency threshold $\tau$ and an upper bound $N$ on the size of the frequent patterns. To compensate the two sources of error discussed in Sect. 3, we set $\tau = 0.7$ as a rule of thumb and $N = 10$ to control the runtime. For the experiments with EEGL, we have carried out three iterations, except for one case. Recall from Sect. 3 that $X$ is set to the matrix of ones. Thus, the first round can be regarded as the vanilla setting for GCN because the feature matrix provides *no* information. Finally, for all experiments with GCN and EEGL, we have used 10-fold cross validation.

## 4.2 EXPERIMENTAL RESULTS AND ANALYSIS

The results for the motifs in Fig. 1 are presented in Table 1. The rows correspond to the feature matrix definitions for GCN and to the three iterations of EEGL. We report the means of the 10 *weighted* F1-scores (percentage) obtained for the 10 folds, together with the standard deviations. As expected, the best result (always close to 100%) is consistently achieved by CGN with the label encoding setting (row LE), which uses the target function in the definition of the feature matrix. Thus, values in row LE should be regarded as *upper bounds* on performance achievable by CGN. Below we give our answers to the three questions formulated in Sect. 1. The quantitative experimental results are accompanied with an *interpretative analysis*. We peek under the hood of EEGL to understand its training dynamics.

**Answer to Q1** In order to answer this question from the introduction, we need to compare the results obtained for the vanilla setting (R0) to those for the second (R1) and third (R2) iteration of EEGL (see Table 1). While the results are inconclusive for $M_1$ and $M_1''$ (there are no sharp differences between $R0$ and $R_1, R_2$), there is a remarkable improvement for $M_1'$. Recall that in case of $M_1'$ (Fig. 1b), the node with label 4 breaks the local symmetry of the house graph, implying that classes 1 and 2 are unions of different orbits. This property holds, at least implicitly, for $M_1$ as well. A closer look at the confusion matrices shows that the cause of the error in R0 is the difficulty to distinguish between base graph nodes and the attachment node of label 4. In contrast to $M_1$ and $M_1''$, there is a sharp improvement already from the second iteration of EEGL in case of the structurally more complex motif pair $M_2$ and its variants $M_2', M_2''$. We illustrate this improvement by the confusion matrices shown in Fig. 2. They were calculated for the test nodes for $M_2'$ in one of the 10 folds. Out of the 94 test nodes in this fold, only 52 nodes have been classified correctly by the vanilla setting (Round-0). Using the maximal frequent patterns extracted from the explanations in the definition of the feature matrix, the number of correctly classified nodes increases to 71 (Round-1), which, in turn, is improved in the next iteration of EEGL to 92 (Round-2), i.e., only two nodes are misclassified. It

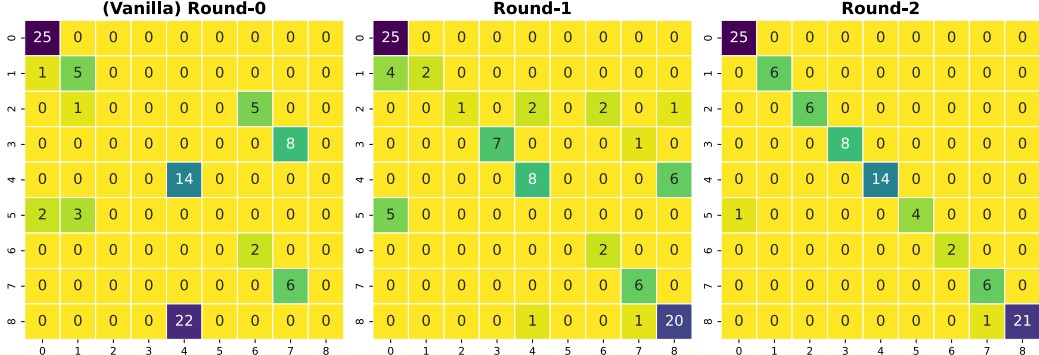

Figure 2: Example of the confusion matrices for a fold of $M_2^{'}$

is important to note that in case of $M_2$, $M_2'$, and $M_2''$, there is a further improvement in R2 compared to R1. This phenomenon cannot be observed for $M_1$ and its variants because for these motifs, the results obtained in the second iteration (R1) are already very close to those with label encoding (LE). This indicates that, at least on structurally more complex motifs, EEGL is capable of *iterative self-improvement* from explanations, with the remark that in case of a few folds, we could observe the opposite, i.e., a drop in the predictive performance.

**Answer to Q2** As noted earlier, LE (label encoding) provides an *upper bound* for achievable prediction, and it achieved almost perfect prediction for each variant. The "adversarial" (A) feature initialization addresses the issue, raised in (8), that any set of subgraph features is expected to improve upon the prediction performance of MPNN. Comparing the improvement of EEGL with other feature initializations seems to be a nontrivial matter. One possibility would be to consider random subgraph features, but it is not clear what would be the relevant notion of a random graph here. We consider a weakest possible formulation, and try to show that EEGL brings more improvement than *some* feature initialization. This, however, includes cases when the subgraph features do not occur at all, and thus feature initialization trivializes. Therefore we attempt to find an "unhelpful" but nontrivial subgraph initialization. The approach is to use subgraph features which do not occur at all in the motifs, but may occur in the base graph. As the base graph is random, the features are expected to be of limited help. Note that motif nodes get a trivial initialization, but the nontrivial initialization of the base graph nodes may have an effect on the motif nodes. Similarly to EEGL, these features are problem dependent as well. The experiments show that indeed, *EEGL is better than A* (see Table 1), and the difference between the two is larger for the larger variants. Randomized feature initialization has been shown to be powerful in a theoretical sense (1; 10). Randomization extends MPNN in a different direction than EEGL. The comparison is inconclusive for the three rounds in Table 1: randomization performs *better* for $M_2''$, but *worse* for all other motifs. However, for six iterations randomization becomes *worse* for this most "complex" pattern $M_2''$ as well (see Table 4.2). The results in Table 4.2 were obtained by *reruning* EEGL on the $M_2''$ datasets, now with six iterations, and by calculating mean and standard deviation of the results of five 10-fold cross-validations for each iteration. In summary, the results on the synthetic datasets show that EEGL outperforms other node feature definitions in predictive performance, with the remark that the number of iterations needed by EEGL may depend on the complexity of the structures behind the target classes.

**Answer to Q3** For answering this question we use frequent patterns extracted from the explanations as well. Consider again the confusion matrices in Fig. 2 computed for a fold for $M_2'$ (see Appendix D for the detailed results on $M_2'$). Regarding the vanilla setting (Round-0), note that 83% (5 out of 6) of the nodes of label 2 are misclassified by label 6, 100% (8 out of 8) of the nodes of class label 3 are classified by 7, 60% (3 out of 5) of the nodes of label 5 are classified by 1, and 100% (22 out of 22) of the nodes with class label 8 are classified mistakenly by class 4. The remaining three misclassifications arise from classifying motif nodes with base graph label 0. These misclassifications confirm that vanilla GNN is unable to distinguish between nodes having the same 1-WL label. Indeed, class label pairs 2 and 6, 3 and 7, 5 and 1, and 8 and 4 have the same 1-WL label. We have a different picture for the second iteration (Round-1 in Fig. 2). Consider, for example, the two class labels 4 and 8. The maximal frequent patterns extracted for $M_2'$ in the first iteration are given in the top row of Fig. 3. (The same fold is used as for the confusion matrices in Fig. 2.) One can check

| R0 | R1 | R2 | R3 | R4 | R5 | R6 |
|---|---|---|---|---|---|---|
| $59.21 \pm 0.95$ | $71.42 \pm 4.68$ | $81.68 \pm 5.54$ | $90.08 \pm 3.89$ | $93.48 \pm 2.49$ | $94.09 \pm 2.10$ | **$96.44 \pm 3.08$** |

Table 2: Average weighted F1-score results in percentage (mean $\pm$ SD) obtained for $M_2''$ for six iterations of EEGL. The result for Round-6 (R6) outperforms $95.23 \pm 0.77$, the result for the random (R) setting.

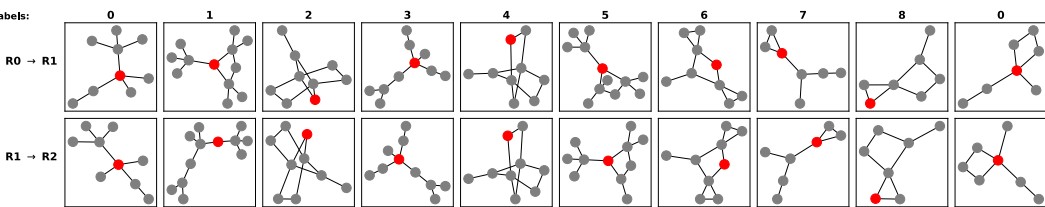

Figure 3: The $d = 10$ maximal frequent subgraphs extracted by EEGL in the first (R0 $\rightarrow$ R1) and the second (R1 $\rightarrow$ R2) iteration for one fold of $M_2'$. Class labels indicated on top. There are two patterns for label 0.

that by removing the node of degree 1 from the maximal frequent pattern extracted from the explanations for label 4, we obtain the entire left-hand side motif in $M_2'$ (see, also, Fig. 1e). Similarly, we obtain a subgraph of the right-hand side motif in $M_2'$, if we remove the node of degree 1 from the maximal frequent pattern extracted for label 8. Thus, the two patterns are genuine in the sense that they *can* distinguish between classes 4 and 8, as they are embedded by rooted (induced) subgraph isomorphism, and *not* by homomorphism. This additional power is reflected in the confusion matrix for iteration Round-1 in Fig. 2. Out of the 36 nodes of label 4 or 8, 22 are misclassified in Round-0, and only 7 in Round-1. After the second iteration the patterns for classes 4 and 8 do not change (see the bottom row of Fig. 3). Still, as shown in the last confusion matrix (see Round-2 in Fig. 2), only 1 node out of 36 remains misclassified, due to the fact that the other patterns may change and can therefore influence the feature matrix. We speculate that the small changes in the patterns for label 1 and label 5 (see Fig. 3) also contribute to this better second prediction accuracy on nodes with label 4 or 8.

**Runtime** EEGL first selects $d' \leq d$ frequent rooted patterns (see line 10 of Alg. 1) and then calculates the feature vector for all nodes by checking rooted induced subgraph isomorphism for each of the $d'$ patterns (line 11). EEGL needed in average between around 20 minutes (motif $M_1$) and 1 hour (motif $M_2''$) for one iteration and for a single fold. Since EEGL computes everything from scratch in each iteration, the runtime grows linearly with $K$ (number of iterations in Alg. 1).

## 5 CONCLUDING REMARKS

We introduced EEGL, an iterative XAI-based model improvement approach to extend MPNN using frequent subgraph mining of explanation subgraphs to obtain features for improving predictive performance. The approach produced encouraging initial results, and it poses many directions for further research. We mention two general questions, continuing the list from the introduction.

Q4: How do predictive performance and running time scale with the complexity of the motifs?

As noted earlier, the bottleneck is feature annotation. We plan to explore several improvement options, such as producing explanations from restricted tractable classes (an interesting challenge in itself, requiring new denoising procedures) and using only a subset of the nodes for annotation.

Q5: Does improved prediction performance in the iterations imply improved explanation quality?

There are several metrics for evaluating GNN explanations (16). The evaluation of explanations in XAI is a difficult issue, with several results indicating the problematic nature of explanations produced (see, e.g. (2)). The lack of ground truth exacerbates the difficulties. Considering synthetic problems where ground truth is available may be useful, but (12) warns of possible pitfalls. Considering node feature explanations could also be useful in this context.

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
