# OpenReview forum: "Iterative Graph Neural Network Enhancement Using Explanations"
_ICLR.cc/2024/Conference — ICLR 2024 Conference Withdrawn Submission_

### Official Review · Reviewer_9PFw · 2023-10-24

**Soundness:** 1 poor
**Presentation:** 1 poor
**Contribution:** 2 fair
**Rating:** 3
**Confidence:** 4

**Summary:**

The paper aims to enhance Graph Neural Networks (GNN) using Explanatory Artificial Intelligence (XAI). Specifically, the EXPLANATION ENHANCED GRAPH LEARNING (EEGL) strategy is introduced for node classification tasks in GNNs. It utilizes frequent connected subgraph mining to identify and analyze patterns in explanatory subgraphs. And iteratively update feature matrix by annotating subgraph information.

**Strengths:**

1. Introducing interpretation into graph neural network architecture is interesting.
2. The research topic is important.

**Weaknesses:**

1. My first concern is the complexity of the model. In every iteration, the approach starts by utilizing a node explainer to derive all explanation graphs. Subsequently, it calculates the maximal frequent rooted subgraphs for all labels. The algorithm then selects the top-k rooted subgraphs by processing them through the GNN and determining the F1-score. Finally, a feature annotation module updates the feature matrix X. Notably, in the experiments, each iteration on one fold takes about 1 hour and 20 minutes. I urge the authors to provide a thorough analysis of their method's complexity and compare its runtime to other baseline models.
2. My second point of contention lies in the experimental setup. The absence of baseline models for comparison in the authors’ work makes it hard  to convincingly showcase the efficacy of their proposed method.
3. My third concern is that the datasets are all Synthetic Data. I highly recommend authors to incorporate more real-world dataset for experiments.
4. The paper's presentation lacks clarity, particularly in section 3 where the notation introduction feels disorganized. I suggest that the authors compartmentalize definitions into distinct sections or blocks for better readability. Furthermore, the method's description relies solely on textual explanations. Incorporating a diagram or figure to illustrate the workflow of the proposed method would enhance understanding.

**Questions:**

Please refer to Weaknesses section

---

> ### Author Response · Authors · 2023-11-16
> **Response**
>
> We thank the Reviewer for their review of our paper. In the response we refer to weaknesses by W.
>
> **Answer to W1:**  In the updated version we are going to provide details.
>
> **Answer to W2:** Thank you very much for this question. We consider it to be a reformulation of Q2 in the introduction. We interpret the question as comparisons to single runs of GNN involving various non-adaptive initialization techniques. Answers are given on page 8.
>
> **Answer to W3:** The choice of using synthetic data is explained in detail in Section 4.1 (page 5), building upon the negative findings of [18] regarding real-world benchmarks for GNN node classification.
>
> **Answer to W4:** Thanks for suggesting compartmentalization. As the notation is standard, we tried to present it as compactly as possible in order to save space. We will introduce subsections.
>
> Algorithm 1 on page 3 contains the pseudocode of the method. As noted on page 3, a diagram is included as Figure 4 in  Appendix B. Both could not be included in the main text due to lack of space.

---

### Official Review · Reviewer_FGzN · 2023-11-01

**Soundness:** 2 fair
**Presentation:** 3 good
**Contribution:** 2 fair
**Rating:** 3
**Confidence:** 5

**Summary:**

This paper considers using explanation to update input graphs iteratively, guiding the model to rely on only identified critical elements for node classification. Particularly, it want to discover frequent subgraph structures, and extend node attributes with indicators of subgraph isomorphism existence.

**Strengths:**

1. It explores using GNN explanations to augment node attributes, and test its potential to improve GNN performance for node classification. It is an interesting and challenging direction
2. It designs EEGL, containing both subgraph pattern extraction module and feature annotation module to iteratively examine explanations and augment node representations. Algorithms are provided for each components.
3. It shows promising results in experiments.

**Weaknesses:**

1. The basic assumption seems incomplete. When GNN models can not go beyond 1-WL algorithm, why augmenting graphs using their explanations can help? Their explanations would be the same for nodes with 1-WL isomorphism.
2. Experiments are incomplete. No comparisons are made with other graph augmentation techniques. No comparisons are made with other explanation-guided learning strategies. And its influence to different GNN architectures are not tested.
3. Time complexity is missing. The pattern detection and node attribution modules may require a lot of time to run. Authors discussed about two techniques used, and talked about running time for one iteration. But the complexity analysis and comparisons are needed for fully understand its computation cost.

**Questions:**

Please refer to the weakness part.

---

> ### Author Response · Authors · 2023-11-16
> **Response**
>
> We thank the Reviewer for their review of our paper. In the response we refer to weaknesses by W.
>
> **Answer to W1:** The explainer returns a rooted subgraph for each node (for example, a triangle). This subgraph is found by a sensitivity-type analysis. It is unrelated to the 1-WL algorithm. Thus explanations for ``non-isomorphic'' nodes that are 1-WL indistinguishable can be different.
>
> **Answer to W2:** Graph augmentation techniques (see Sec. 5.2 in Morris et al., 2021) use subgraph augmentation for graph classification problems. The only exception we are aware of is the paper by Zeng et al. (2022). We plan to make a comparison to this technique. As far as we know, explanation-guided GNN learning has not been considered before. Our approach can be used with any GNN learner and explainer. The detailed study of the effect of using different systems requires a separate paper, which is in progress.
>
> **Answer to W3:** The Reviewer is right, this was indeed the case. In the updated version we are going to provide details.

---

### Official Review · Reviewer_dVAw · 2023-11-06

**Soundness:** 1 poor
**Presentation:** 1 poor
**Contribution:** 2 fair
**Rating:** 3
**Confidence:** 4

**Summary:**

This work proposes a framework, EEGL, to enhance graph neural networks by incorporating the explanations. The effectiveness of the proposed work has been demonstrated on the synthetic datasets.

**Strengths:**

The idea of using explanation to enhance graph neural networks is convincing.

**Weaknesses:**

1.	The paper demands a meticulous review by a native English speaker, to refine its clarity and presentation. While numerous areas require attention, highlighted below are some specific instances. It should be noted that this isn't an exhaustive list:
a.	It will be better to provide the full term before transitioning to an abbreviation. For instance, before using 'XAI', its complete form should be mentioned.
b.	In the abstract, the statement “EEGL is an iterative algorithm….. in the node neighborhoods” is too long to understand. Following that, the sentence “Giving an application-dependent algorithm for such an extension of the Weisfeiler-Leman (1-WL) algorithm has been posed as an open problem.” appears disjointed from the context and needs rephrasing for clarity.
c.	The term 'MPNN' typically stands for "message passing neural networks." Instead of “Message-passing GNN”.
d.	The commonly-used abbreviation for graph neural networks is 'GNNs.' If the paper opts to use 'GNN' as the abbreviation, phrasing like "GNN form" in the introduction should be revised to "The GNN forms.”
e.	In Figure 1, the meaning of designations such as “M1, M1’, M1’’, M2, M2’, M2’’” should be elucidated within the caption.
f.	The current citations format does not adhere to the official template provided.
2.	The exclusive reliance on synthetic datasets without any inclusion of real-world datasets diminishes the empirical strength of the paper. While the authors have proffered reasons for this choice, the graph generation model utilized is rather simple. Incorporating a more enhanced model, such as Stochastic Block Model, might enhance the quality and representativeness of the generated graphs, allowing for a more robust validation of the proposed method.

In summary, while the paper holds potential, its current presentation hinders a comprehensive review. I'm eager to provide a more detailed review if the authors could provide an updated version.

**Questions:**

See the weaknesses

---

> ### Author Response · Authors · 2023-11-16
> **Response**
>
> We thank the Reviewer for their review of our paper. In the response we refer to weaknesses by W.
>
> **Answer to W1:** Weaknesses a)-f) are all minor notation, terminology and style issues, which will be addressed in the updated version.
>
> **Answer to W2:** A detailed justification for considering synthetic data is given in the first paragraph of Sect. 4.1. The particular model we use is a variant which has been considered in many forms since its introduction in (Ying et al., 2019). Increasing motif complexity, the number of motifs, and the graph-theoretical properties of the labeling can make it very complex and challenging. Our experiments are about exploring this challenge. On the other hand, models, such as the stochastic block model, are useful
> when the underlying graph serves as a similarity relation (homophily, heterophily) for the node feature vectors, but not for node classification tasks, where the class labels are determined by graph patterns. Thus the stochastic bloc model is not relevant in our context. The working assumptions discussed in the first paragraphs on p. 2 imply that our focus is orthogonal to this situation.
>
> **It is not clear why the apparently minor issues listed in Weakness 1 lead to the Reviewer's withholding their more detailed review.**

---

### Official Review · Reviewer_CfEt · 2023-11-09

**Soundness:** 4 excellent
**Presentation:** 3 good
**Contribution:** 3 good
**Rating:** 6
**Confidence:** 4

**Summary:**

This paper proposes a new XAI-based iterative approach called Explanation Enhanced Graph Learning (EEGL) to enhance the performance of node classification by focusing on explanation subgraph structures. The proposed method applies frequent subgraph mining to explanation graphs to find helpful patterns in each class. Then, discovered patterns are used to annotate the feature matrix for GNN. Experiments show that the proposed algorithm can improve the performance on various synthetic data that the 1-WL label cannot distinguish.

**Strengths:**

- The proposed algorithm EEGN is a novel approach that uses frequent subgraph mining to XAI of GNNs for node classification.

- EEGL can improve the performance of node classification by iteratively annotating the information of explanation subgraphs obtained by frequent subgraph mining into a feature matrix.

- This paper empirically shows the effect of feature initialization of GNNs by comparing randomly assigned labels and labels obtained from EEGL-GNN, which can predict labels of synthetic datasets that 1-WL cannot distinguish.

**Weaknesses:**

- It requires additional computational cost to update the feature matrix in larger graphs when finding induced subgraphs. This additional cost can be high, hence it should be empirically evaluated.

- In the pattern extraction module, hyperparameters are determined by the rule of thumb. It is unclear how sensitive the frequent threshold $\tau$ and an upper bound $N$ are in frequent subgraph mining and how much subgraph structures affect the predictive performance.

**Questions:**

- Does R0 means that the feature matrix is initialized with one? It seems that this initialization causes the fact that GCN cannot train well and predict a label as an almost random class in the M2 dataset. Is it due to ill-initialization or the nature of the 1-WL algorithm of GNN?

- As the number of iterations increases, does the annotated feature matrix X converge to that equivalent to the label encoder, and the subgraph structures obtained from GNNExpiner will be equivalent to explanation subgraphs when initialized by the feature matrix with the label encoder?

- There are some typos in the paper, such as in Sec 4.2, CGN (GCN?), the subscript of R, Table 4.2 (no need), and the label in Figure 3.

---

> ### Author Response · Authors · 2023-11-16
> **Response**
>
> We thank the Reviewer for their review of our paper.
> In the response we refer to weaknesses by W, and questions by Q.
>
> **Answer to W1:**
> The Reviewer is right, this was indeed the case. In the updated version we are going to provide details.
>
> **Answer to W2:** The effect of subgraph structures has been the focus of our experiments in the current paper, and Section 4.2 contains several qualitative observations beyond the quantitative results. The choice of the graph mining hyperparameters may be application-dependent and it would require detailed further study.
>
> **Answer to Q1:** Thank you very much for this valid question. Regarding your first question, the answer is yes. This is the vanilla setting in the sense that (effectively) no node feature is used in GCN. For this case, the poor predictive performance of GNN on 1-WL-indistinguishable nodes follows from its inherent limitation.
> Thus the answer to the second question is that it is the nature of 1-WL.
> (In fact, EEGL iteratively turns an ill initialization into a non-ill one.)
> As an example, consider motif $M_2'$ (cf. Fig.~1). Note that vanilla GCN correctly classifies all nodes of labels 4 (cf. the confusion matrix for Round-0 in Fig. 2) and that all other nodes are classified with a label different from 4, except for the 22 nodes with label 8. The reason is that although nodes with label 4 and 8 are probably 1-WL distinguishable in the entire graph due to the presence of the base graph, they are hard to distinguish for GNN. We recall that the limitations of the distinguishing power of 1-WL provide only an upper bound on that of GNN.
>
> **Answer to Q2:** Thank you very much for this great question! This is exactly what we plan to do in future work described as Q5 at the very end of the paper. As discussed in that paragraph, there are several interesting challenges in this direction, starting with defining the proper metric (some kind of graph distance) for convergence.
>
> **Answer to Q3:** They will be fixed in the updated version.